# Nanosecond anomaly detection with decision trees and real-time application to exotic Higgs decays

S. T. Roche [1,2], Q. Bayer [2], B. T. Carlson [2,3], W. C. Ouligian[2], P. Serhiayenka[2], J. Stelzer [2] & T. M. Hong [2] ✉

We present an interpretable implementation of the autoencoding algorithm, used as an anomaly detector, built with a forest of deep decision trees on FPGA, field programmable gate arrays. Scenarios at the Large Hadron Collider at CERN are considered, for which the autoencoder is trained using known physical processes of the Standard Model. The design is then deployed in real-time trigger systems for anomaly detection of unknown physical processes, such as the detection of rare exotic decays of the Higgs boson. The inference is made with a latency value of 30 ns at percent-level resource usage using the Xilinx Virtex UltraScale+ VU9P FPGA. Our method offers anomaly detection at low latency values for edge AI users with resource constraints.

Unsupervised artificial intelligence (AI) algorithms enable signal-agnostic searches beyond the Standard Model (BSM) physics at the Large Hadron Collider (LHC) at CERN[1]. The LHC is the highest energy proton and heavy ion collider that is designed to discover the Higgs boson[2,3] and study its properties[4,5] as well as to probe the unknown and undiscovered BSM physics (see, e.g.,[6–8]). Due to the lack of signs of BSM in the collected data despite the plethora of searches conducted at the LHC, dedicated studies look for rare BSM events that are even more difficult to parse among the mountain of ordinary Standard Model processes[9–13]. An active area of AI research in high energy physics is in using autoencoders for anomaly detection, much of which provides methods to find rare and unanticipated BSM physics. Much of the existing literature, mostly using neural network-based approaches, focuses on identifying BSM physics in already collected data[14–70]. Such ideas have started to produce experimental results on the analysis of data collected at the LHC[71–74]. A related but separate endeavor, which is the subject of this paper, is enabling the identification of rare and anomalous data on the real-time trigger path for more detailed investigation offline.

The LHC offers an environment with an abundance of data at a 40 MHz collision rate, corresponding to the 25 ns time period between successive collisions. The real-time trigger path of the ATLAS and CMS experiments[75,76], e.g., processes data using custom electronics using field programmable gate arrays (FPGA) followed by software trigger algorithms executed on a computing farm. The first-level FPGA portion of the trigger system accepts between 100 kHz to 1 MHz of collisions, discarding the remaining ≈ 99% of the collisions. Therefore, it is essential to discovery that the FPGA-based trigger system is capable of triggering potential BSM events. A previous study aimed at LHC data has shown that an anomaly detector based on neural networks can be implemented on FPGA with latency values between 80 to 1480 ns, depending on the design[77].

In this paper, we present an interpretable implementation of an autoencoder using deep decision trees that make inferences in 30 ns. As discussed previously[78,79], decision tree designs depend only on threshold comparisons resulting in fast and efficient FPGA implementation with minimal reliance on digital signal processors. We train the autoencoder on known Standard Model (SM) processes to help trigger the rare events that may include BSM.

In scenarios for which a specific BSM model is targeted and its dynamics are known, dedicated supervised training against the SM sample, i.e., BSM-vs-SM classification, would likely outperform an unsupervised approach of SM-only training. The physics scenarios considered in this paper are examples to demonstrate that our auto-encoder is able to trigger on BSM scenarios as anomalies without this prior knowledge of the BSM specifics. Nevertheless, we consider a benchmark where our autoencoder outperforms the existing conventional cut-based algorithms.

[1]School of Medicine, Saint Louis University, Saint Louis, MO, USA. [2]Department of Physics and Astronomy, University of Pittsburgh, Pittsburgh, PA, USA. [3]Department of Physics and Engineering, Westmont College, Santa Barbara, CA, USA. ✉e-mail: tmhong@pitt.edu

Our focus is to search for Higgs bosons decaying to a pair of BSM pseudoscalars with a lack of sensitivity due to a bottleneck in the triggering step. We examine the scenario in which one pseudoscalar with $m_a = 10$ GeV subsequently decays to a pair of photons and the second pseudoscalar with a larger mass decays to a pair of hadronic jets, i.e., $H \rightarrow aa' \rightarrow \gamma\gamma jj$[80], one of the channels of the so-called exotic Higgs decays[81]. The recent result for this final state[82] does not probe the phase space corresponding to $m_a < 20$ GeV due to a bottleneck from the trigger. The study presented here considers various general experimental aspects of the ATLAS and CMS experiments to show that our tool may benefit ATLAS, CMS, and other physics programs generally. We demonstrate that the use of our autoencoder can increase signal acceptance in this region with a minimal addition to the overall trigger bandwidth.

Beyond our benchmark study, we consider an existing dataset with a range of different BSM models, referred to here as the LHC physics dataset[83], to compare our tool with the results of the previously mentioned neural network-based autoencoder designed for FPGA[77]. Lastly, the robustness of our general method is considered by training with samples having varying levels of signal contamination.

This paper uses Higgs bosons to explore the unknown using real-time computing. But more generally, such inferences made on edge AI may be of interest in other experimental setups and situations with resource constraints and latency requirements. It may also be of interest in situations in which interpretability is desirable[84].

# Results

We describe the design of a decision tree-based autoencoder and the training methodology. We then present our benchmark results of a scenario in which an anomaly detector could trigger on BSM exotic Higgs decays in the real-time trigger path. As a test case, we also consider the LHC physics dataset[83] with which our results are compared using a neural network implementation[77]. Lastly, a study showing our autoencoder's effectiveness to signal contamination of training data is presented.

## Autoencoder as anomaly detector

Our autoencoder (AE) is related to, and extends beyond, those based on random forests[85,86]. We note that there are related concepts in the literature with various levels of algorithmic sophistication[87–90], but these approaches may be more challenging to implement on the FPGA. We build on the deep decision tree architecture that uses parallel decision paths of `fwXmachina`[78,79]. A general discussion of the tree-based autoencoder is given below. The subsections that follow will detail the ML training, the firmware design, including verification and validation, and the simulation samples.

A tree of maximum depth $D$ takes an input vector $\mathbf{x}$, encodes it to the latent space as $\mathbf{w}$, then decodes $\mathbf{w}$ to an output vector $\hat{\mathbf{x}}$. Typically both $\mathbf{x}$ and $\hat{\mathbf{x}}$ are elements of $\mathbb{R}^V$ while $\mathbf{w}$ is an element of $\mathbb{R}^T$, where $V$ is the number of input variables and $T$ is the number of trees, i.e.,

$$\mathbf{x} \overset{\overbrace{\overset{\text{autoencoder}}{\xrightarrow{\text{encoder}}} \mathbf{w} \xrightarrow{\text{decoder}}}}{} \hat{\mathbf{x}}. \tag{1}$$

Typically the latent space is smaller than the input-output space, i.e., $T < V$, but it is not a requirement. A decision tree divides up the input space $\mathbb{R}^V$ into a set of partitions $\{P_b\}$ labeled by bin number $b$. The $b$ is a $B$-bit integer, where $B \leq 2^D$, since the tree is a sequence of binary splits.

The encoding occurs when the decision tree processes an input vector $\mathbf{x}$ to place it into one of the partitions labeled by $w$. If more than one tree is used, then $w$ generalizes to a vector $\mathbf{w}$. The decoding occurs when $\mathbf{w}$ produces $\hat{\mathbf{x}}$ using the same forest. The bin number $b$

corresponds to a partition in $\mathbb{R}^V$, which is a hyperrectangle $P_b$ defined by a set of extrema in $V$ dimensions.

A metric $d$ provides an anomaly score calculated as a distance between the input and output, $\Delta = d(\mathbf{x}, \hat{\mathbf{x}})$, which is our analog of the loss function used in neural network-based approaches. Our choice for the estimator of $P_b$ is the dimension-wise central tendency of the training data sample in the considered bin, $\hat{\mathbf{x}} = \text{median}(\{\mathbf{x}\}) \forall \mathbf{x} \in P_b$. The median minimizes the $L^1$ norm, or Manhattan distance, with respect to input data resembling the training sample.

The encoding and decoding are conceptually two steps, with the latent space separating the two. But, as explained in the next section, our design executes both steps simultaneously and bypasses the latent space altogether by a process we call ⋆coder (star-coder), i.e., $\hat{\mathbf{x}} = \star \mathbf{x}$,

$$\mathbf{x} \overset{\star\text{coder}}{\longrightarrow} \hat{\mathbf{x}}. \tag{2}$$

Finally, the anomaly score is the sum of the $L^1$ distances for each tree in the forest, i.e.,

$$\Delta(\mathbf{x}) = d(\mathbf{x}, \star \mathbf{x}) = \sum_{\substack{\text{trees} \\ t}} \sum_{\substack{\text{vars} \\ v}} |x_v - \star x_{v,t}|. \tag{3}$$

When the parameters of the autoencoder are trained on known SM events, the autoencoder ideally produces a relatively small $\Delta$ when it encounters an SM event and a relatively large $\Delta$ when it encounters a BSM event. The metric sums the individual distances for variables of different types, such as angles and momenta, so the ranges of each variable must be carefully considered. At the LHC they are naturally defined by the physical constraints, e.g., 0 to $2\pi$ for angles and 0 to $p_T^{max}$, the kinematic endpoint, for momenta. The values are transformed into binary bits to design the firmware; see Appendix C.3 of Ref. 78 for a detailed discussion.

An illustrative example of the decision tree structure is given in Supplementary Fig. 1, and a demonstration of the autoencoder using the MNIST dataset[91] is given in Supplementary Fig. 2.

## ML training

The machine learning (ML) training of the autoencoder described here is novel and is suitable for the physics problems at hand. Qualitatively, the training puts small-sized bins around regions with high event density and large-sized bins around regions of sparse event density. An illustration of the bin sizes is given with a 2d toy example in Supplementary Fig. 3, which shows the decreasing sizes of bins as the tree depth increases.

The following steps are executed. To start, $\mathbf{x} = \{x_v\} = \{x_0, x_1, ..., x_{V-1}\}$ is a vector of length $V$, the number of input variables, that describes the training sample $S$. (1) Initialize $s$ with $S$ in steps 2–4 and depth $d = 1$. (2) For the sample $s$, the PDF $p_v$ is the marginal distribution of bit-integer-valued input variable $x_v$ for a given $v$. The PDF $p_m$ is the distribution of the maximum values of the set $\{p_v\}$. Sampling the maximum-weighted PDF $m \cdot p_m$ gives $\tilde{m} = m_{\tilde{v}}$ that corresponds to the $x_{\tilde{v}}$. (3) The PDF $p_{\tilde{v}}$ is for the $x_{\tilde{v}}$ under consideration. Sampling $p_{\tilde{v}}$ yields a threshold value $\tilde{c}$. (4) The sample $s$ is split by a cut $g = (x_{\tilde{v}} < \tilde{c})$. (5) The steps 2–4 are continued recursively for the two subsamples until one of two stopping conditions are met: (condition-i) the number of splits exceeds the maximum allowed depth $D$, (condition-ii) the split in step 3 produces a sample that is below the smallest allowed fraction $f$ of $S$. (6) When stopped, the procedure breaks out of the recursion by appending the requirement $g$ to the set $G$. (7) In the end, the algorithm produces a partition $G$ of the training sample called the decision tree grid (DTG) that corresponds to a deep decision tree (DDT) illustrated in Fig. 1. The pseudocode given below finds $G = \text{DTG}(S, \emptyset, 1)$.

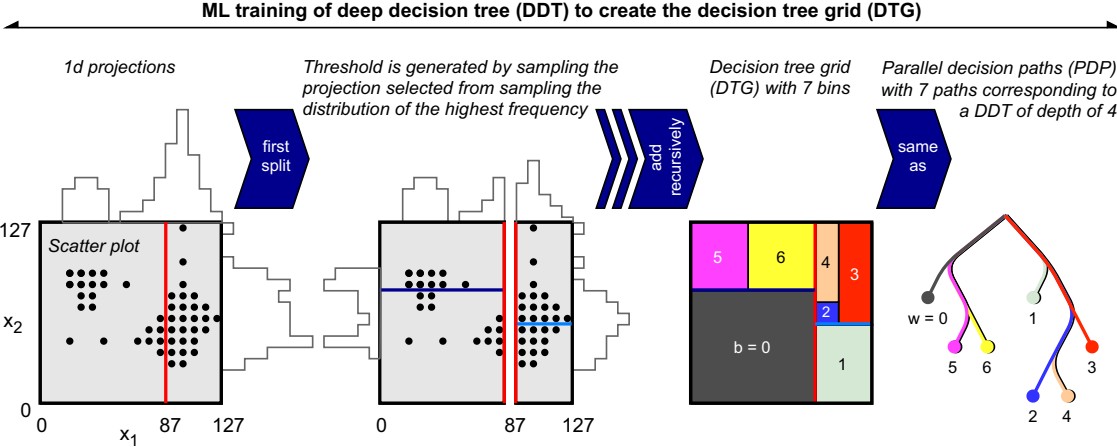

**Fig. 1 | Illustration of the ML training.** Data is represented as $x_1$ vs. $x_2$ (leftmost). Recursive importance sampling considers the marginalized distributions (second). A decision tree grid is constructed (third). Deep decision trees with maximum depth of 4 corresponds to parallel decision paths (rightmost).

**function** DTG(training sample $s$, partition $G$, depth $d$)
1: if $(|s|/|S| < f$ or $d > D)$, then
2: return $G$
3: end if

4: $p_v \leftarrow \text{PDF}(x_v) \ \forall \ x_v \in \mathbf{x}$
 • Identify the variable $x_{\tilde{v}}$ to cut on
 Build set of pdfs for input variables
5: $p_m \leftarrow \text{PDF}(\{\max(p_v)\} \forall \ v \in V)$ Build pdf of max of input pdfs
6: $\tilde{m} \leftarrow \text{sample}(m \cdot p_m)$ Sample max-weighted pdf
7: $\tilde{v} \leftarrow v$ where $m_v = \tilde{m}$ Find variable index
 • Find threshold $\tilde{t}$ to cut on $x_{\tilde{v}}$
8: $\tilde{c} \leftarrow \text{sample}(p_{\tilde{v}})$ Sample variable pdf
9: $g \leftarrow x_{\tilde{v}} < \tilde{c}$ Make selection
 • Build partition
10: $G \leftarrow$ append $g$ Add to $G$ the new selection $g$
 • Recursively build the decision tree
11: call DTG($s$ if $g, g, d + 1$) Call DTG on subset passing $g$
12: call DTG($s$ if not $g$, not $g, d + 1$) Call DTG on subset failing $g$
13: return $G$

Weighted randomness in both variable selection $x_{\tilde{v}}$ and threshold selection $\tilde{c}$ allow for the construction of a forest of non-identical decision trees to provide better accuracy in the aggregate. As our ML training is agnostic to the signal process, the so-called boost weights are not relevant because misclassification does not occur in one-sample training.

An information bottleneck may exist, where the input data is compressed in the latent layer of a given autoencoder design, then subsequently decompressed for the output. For our design, the latent layer is the output of the set of decision trees $T$ in the forest. Accordingly, the latent data is the set of bin numbers from each decision tree, i.e., $\{b_0, b_1, \ldots, b_{T-1}\}$. Compression occurs if $T$ is smaller than the number of input variables $V$, i.e., $T/V < 1$. We will see later that the benchmark physics process is not compressed with $T/V$ of about four, while the LHC Physics problem is compressed with $T/V$ of about half. This demonstrates that the autoencoder does not necessarily rely on the information bottleneck but rather on the density estimation of the feature space.

**Simulated training and testing samples**
The training and testing samples are generated using the Monte Carlo method that is standard practice in high energy physics. In our study, we use offline quantities for physics objects to approximate the input values provided at the trigger level, as offline-like reconstruction will be available after the High Luminosity LHC (HL-LHC) upgrade of the level-1 trigger systems of the experiments[92,93]. A brief summary of the samples is given below (see "Methods" for technical details).

The training sample consists of half a million simulated proton-proton collision events at 13 TeV. It is comprised of a cocktail of SM processes that produce a $\gamma\gamma jj$ final state, where $j$ represents light flavor hadronic jets, weighted according to the SM cross sections.

The testing is done on half a million of the above process as the background sample as well as on a signal sample for the benchmark of the Higgs decay process $H_{125} \rightarrow a_{10} a_{70} \rightarrow \gamma\gamma jj$ with asymmetric pseudoscalar masses of 10 and 70 GeV, respectively. To show that our training is more generally applicable to other signal models beyond the benchmark, we consider an alternate cross-check scenario with a Higgs-like scalar of a smaller mass at 70 GeV, $H_{70} \rightarrow a_5 a_{50} \rightarrow \gamma\gamma jj$, decaying to pseudoscalars with masses of 5 and 50 GeV, respectively.

The benchmark and the alternate cross-check sample consists of 100 k events each. The $H_{125}$ and $H_{70}$ bosons are produced by gluon-gluon fusion. MadGraph5_aMC 2.9.5 is used for event generation at leading order[94]. Decay and showers are done with Pythia8[95]. Detector simulation and event reconstruction are done with Delphes 3.5.0[96,97] using the CMS card[98].

The input variables to the autoencoder depend only on the two photons and the two jets. The photons are denoted as $\gamma_1$ and $\gamma_2$, which are the two photons with the highest momenta transverse to the beam direction ($p_T$) in the event. Similarly, the two leading jets are denoted as $j_1$ and $j_2$. Photons are reconstructed in Delphes with a minimum $p_T$ of 0.5 GeV. Jets are reconstructed with the anti-$k_t$ algorithm with a minimum $p_T$ of 20 GeV. The input variables to the autoencoder include the $p_T$ of these four objects, along with invariant masses of the diphoton ($m_{\gamma\gamma}$) and dijet ($m_{jj}$) subsystems, and the Cartesian $\eta$-$\phi$ distance ($\Delta R$), where $\eta$ is the pseudorapidity variable defined using polar angle $\theta$ and $\phi$ is the azimuthal angle.

The input variable distributions for the full list of eight variables—$p_T^{\gamma 1}, p_T^{\gamma 2}, p_T^{j1}, p_T^{j2}, \Delta R_{\gamma\gamma}, \Delta R_{jj}, m_{\gamma\gamma}, m_{jj}$—are shown in five plots with white background in Fig. 2. The left-most plots show the $p_T$ distribution for the jets and photons, along with the cuts imposed in Delphes for object reconstruction. The middle column plots show the $m_{jj}$ and two $\Delta R$ distributions; the $\Delta R_{jj}$ distribution shows a peak at $\pi$ for SM processes, which reveals the back-to-back signature in the azimuthal $\phi$ coordinate of the dijet system with respect to the beam direction. The top-right plot shows the $m_{\gamma\gamma}$ distribution with the pre-selection requirement discussed in the next section; the peak at 10 GeV for $H_{125}$ corresponds to the $a_{10}$ in the intermediate state. The bottom-right plot with the shaded background shows the $m_{\gamma\gamma}$ distribution after a cut on the anomaly score from the autoencoder, which is described in the next section.

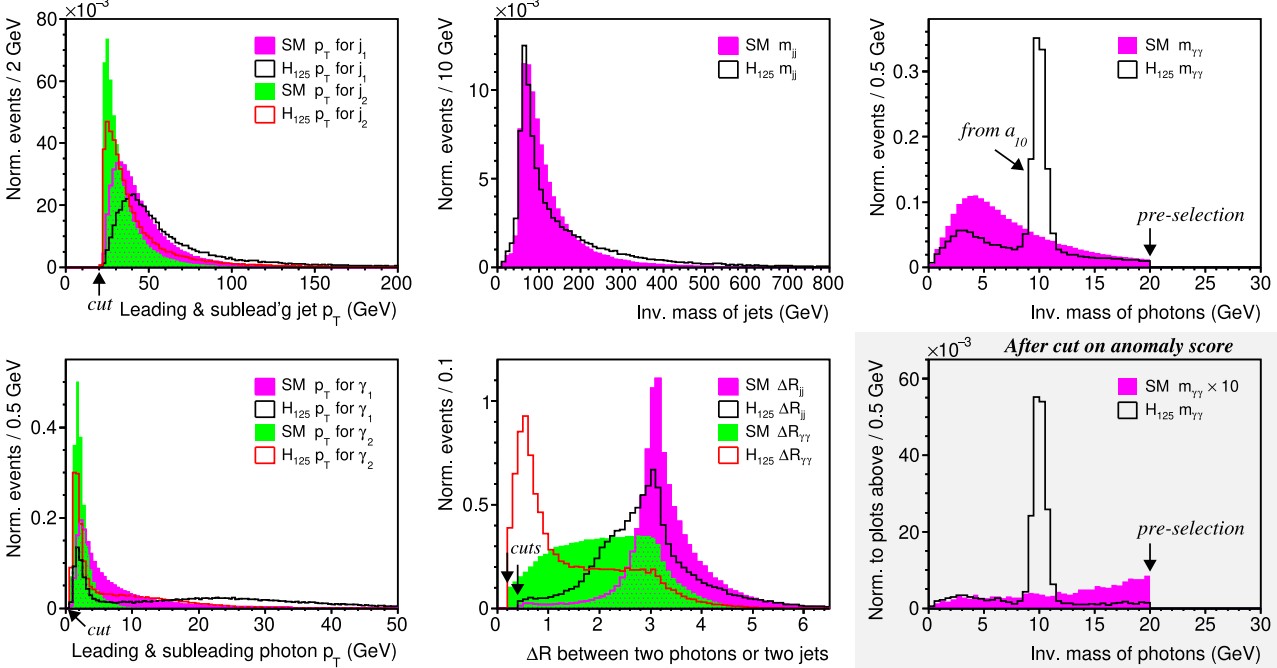

**Fig. 2 |** Input variable distributions for $H_{125} \to a_{10}a_{70} \to \gamma\gamma jj$ and SM $\gamma\gamma jj$ showing (top-left) $p_T$ for the leading and subleading jet, (top-middle) $m_{jj}$ for the dijet subsystem, (top-right) $m_{\gamma\gamma}$ for the diphoton subsystem, (bottom-left) $p_T$ for the leading and subleading photon, and (bottom-middle) $\Delta R$ distance for the dijet and diphoton subsystem. The shaded panel (bottom-right) is the $m_{\gamma\gamma}$ distribution after a cut on the anomaly score of the autoencoder; this plot is normalized relative to the top-right plot before the cut.

## Benchmark: Exotic Higgs decays

In order to define and quantify the gain using the autoencoder trigger in the FPGA-based systems over conventional approaches, we consider the threshold-based algorithm typically deployed at the LHC, such as at the ATLAS and CMS experiments. The most recent analysis of the $\gamma\gamma jj$ final state[82] used the diphoton ($\gamma\gamma$) trigger, so we take this to be representative of the conventional approach. Moreover, as trigger performance is generally comparable between the ATLAS and CMS experiments, we take the ATLAS results from the Run-2 data-taking period (2015–2018) as typical of the situation at the LHC. ATLAS reports a peak event rate of 3 kHz for a diphoton trigger in the FPGA-based first level trigger system in 2018 out of a peak total rate of about 90 kHz[99]. The threshold is $p_T > 20$ GeV for each photon at the first level trigger, but the refined threshold is 35 and 25 GeV for the leading and subleading photon, respectively, in the subsequent CPU-based high-level trigger[100]. The high-level values are more representative of the thresholds for which the first-level trigger becomes fully efficient, so we approximate the situation by requiring 25 GeV for each of the two reconstructed photons. We consider this to be the ATLAS-inspired cut-based diphoton trigger.

The events of interest containing $\gamma\gamma jj$ constitute a subset of all events that pass the diphoton requirement, as $\gamma\gamma$ events accompanied with zero or one jet ($\gamma\gamma$ or $\gamma\gamma j$, respectively) would also pass. However, determining the precise composition of the events passing the diphoton trigger is a nontrivial task. So for our comparisons below, we consider the worst-case scenario to assume that the $\gamma\gamma jj$ event rate equals the entire event rate of the diphoton trigger. It is considered the worst-case scenario because the more likely case that the $\gamma\gamma jj$ rate is less than the $\gamma\gamma$ rate would give a more favorable result for the autoencoder in comparison.

The overall rate is estimated by comparing the fraction of the $\gamma\gamma jj$ simulated background sample accepted by the autoencoder with the diphoton trigger, which has a known event rate. The SM processes that contribute to this trigger rate have been studied using a procedure similar to the one we describe[101]. The study identifies two dominant

scenarios that yield two reconstructed photons: (1) the SM process in which $\gamma\gamma$ originate from the interaction vertex and (2) the SM process in which one photon is accompanied by a jet that has photon-like characteristics ($\gamma j$). The study shows that the shape of the $m_{\gamma\gamma}$ distribution for events from the $\gamma\gamma$ process and $\gamma j$ are similar. Therefore, we conclude that a comparison of equal acceptance using a sample dominated by the $\gamma\gamma$ is a conservative approximation for the totality of these SM processes, comprised of both $\gamma\gamma$ and $\gamma j$, corresponding to the above-mentioned 3 kHz.

The diphoton trigger performance is approximated by applying the $p_T^{\gamma 2} > 25$ GeV threshold, as discussed above, to the subleading reconstructed photon in the simulated sample described in the previous section. Compared to the previous results[82], we note that a non-negligible amount of $H_{125}$ passes the diphoton trigger in this study in the $m_a < 20$ GeV region because we are assuming an offline-like reconstruction after the HL-LHC upgrade of the level-1 trigger systems of the experiments[92,93]. In the SM sample, 0.31% of events passed this ATLAS-inspired diphoton trigger. For the benchmark Higgs $H_{125}$ decay, 2.2% of the events passed. For the alternate cross-check $H_{70}$ decay, 0.01% passed; the small acceptance is due to the soft photon spectrum from the $a_5$ decay.

The autoencoder trigger performance is evaluated after the following pre-selection. In both training and testing, the autoencoder is exposed only to events that (1) have two or more reconstructed photons and two or more reconstructed jets and (2) have two photons that fall within the previously unexamined range $m_{\gamma\gamma} < 20$ GeV. Events that do not meet these requirements are discarded. A total of 38% of the SM background sample pass the pre-selection, as did 53% of the $H_{125}$ sample and 29% of the $H_{70}$ sample.

The autoencoder is trained using a forest of 30 decision trees at a maximum depth of 6 on the training sample of the SM process. In the training step, measured quantities corresponding to the offline reconstruction of physics objects are used as input variables. The trained autoencoder model is applied to both the testing sample of the SM considered as the background process and the benchmark

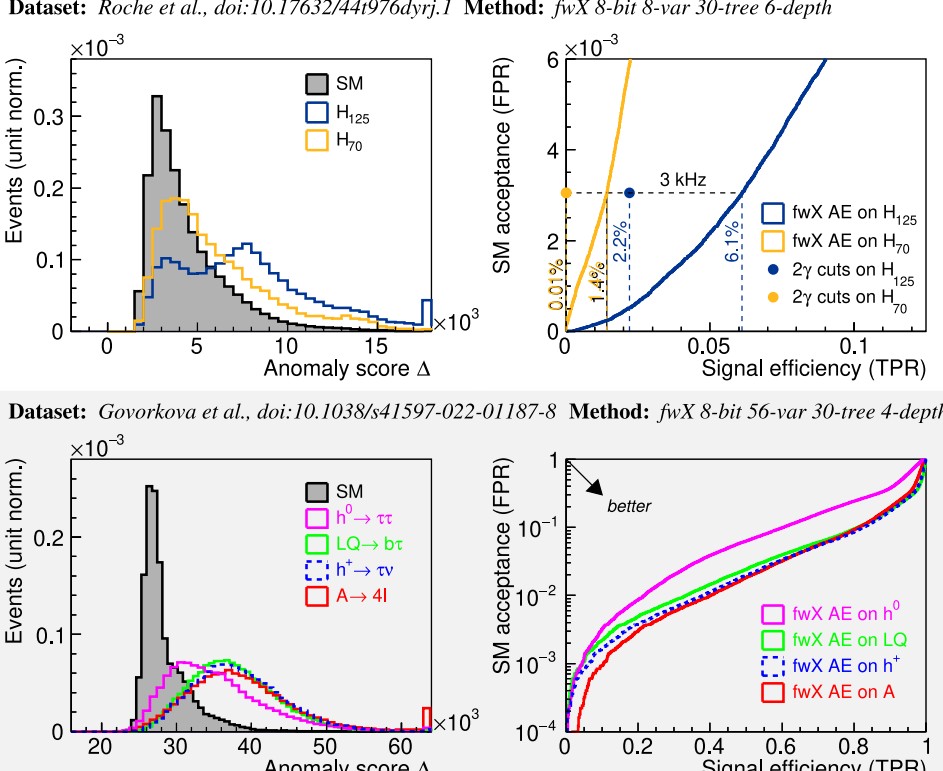

**Fig. 3 | Physics performance results.** The distribution is given for anomaly scores $\Delta$ (left column) and the ROC curves (right column) for the $H \to aa' \to \gamma\gamma jj$ scenario (top row) and the LHC physics dataset[83] (bottom row). Along with the ROC curves for the $\gamma\gamma jj$ dataset (top right), the operating points of the $p_T^{\gamma 2} > 25$ GeV trigger are shown, with numerical values to compare it to the autoencoder's performance. Values shown are fractions of all events in the sample. The autoencoder is trained only on the respective Standard Model ($SM_{\gamma\gamma jj}$ and $SM_{cocktail}$) processes. TPR and FPR represent true and false positive rates, respectively. The plots are software-simulated results using bit integers as done in the firmware.

$H_{125}$ sample as the signal process. In the evaluation step, offline quantities are converted to bitwise values to mimic the firmware[78]. The cross-check $H_{70}$ sample is also considered an alternate signal process to demonstrate that the autoencoder is effective over a wide kinematic range.

Anomaly scores for each event are calculated, and their distributions are shown in the top-left plot of Fig. 3. The corresponding ROC curves are shown on the top-right plot in the same figure. The plots in the bottom row are for a different physics scenario, which is discussed in the next section.

The autoencoder trigger achieves 6.1% acceptance for the benchmark $H_{125}$ signal at the 3 kHz SM rate, nearly triple the 2.2% value using the diphoton trigger. Similarly, the acceptance of the cross-check $H_{70}$ sample is 1.4%, drastically increased from the negligible value of the diphoton trigger at 0.01% for the same rate.

For the FPGA cost, the configuration is run on an Xilinx Virtex UltraScale+ FPGA VCU118 Evaluation Kit (with FPGA model xcvu9p) with a clock speed of 200 MHz. Algorithm latency is 10 clock ticks (30 ns), and the interval is 1 clock tick (5 ns). About 7% of available look-up tables (LUT) are used; 1% of flip flops (FF) are used; a negligible number of digital signal processors (DSP) is used; no BRAM or URAM is used. The results are summarized in the first column of Table 1.

**Comparison: LHC physics dataset**
Our autoencoder is applied to the LHC physics dataset[83] and compared to the results of the neural network implementation[77] that involves discrimination of several different BSM signals from a mixture of SM background. In this dataset, all events include the existence of an electron with momentum transverse to the beam axis $p_T > 23$ GeV and pseudorapidity $|\eta| < 3.0$ or a muon with $p_T > 23$ GeV and $|\eta| < 2.1$. This preselection is designed to limit the data to events that would already pass a real-time single-lepton trigger. We note that this requirement limits the ability of the study to be generalized for events that do not pass an existing real-time algorithm.

The background is composed of a cocktail of Standard Model processes ($SM_{cocktail}$) that would pass the above-mentioned preselection composed of $W \to \ell\nu$, $Z \to \ell\ell$, $t\bar{t}$, and QCD multijet in proportions similar to that of $pp$ collisions at the LHC. The dataset's features are 56 variables consisting of sets of ($p_T$, $\eta$, $\phi$) from the 10 leading hadronic jets, 4 leading electrons, and 4 leading muons, along with $E_T^{miss}$ and its $\phi$ orientation. A cross-check using only 26 of these training variables is presented later in the section.

In our training, a forest of 30 trees at a maximum depth of 4 is trained on a training set of the SM cocktail and evaluated on both a testing portion of the SM cocktail and each of the BSM samples. As the plots in the bottom row of Fig. 3 show, the anomaly detector is able to isolate all signal samples from the background. The areas under the ROC curves (AUC) demonstrate comparable performance. For TPR-FPR convention chosen in Fig. 3, the area under the curve in the plot corresponds to $1 - $ AUC, i.e., an AUC of 1 is an ideal classifier. Our AUC values are listed for the four signal scenarios and neural network-based results for DNN VAE PTQ 8-bit, the configuration highlighted in Ref. 77, in parentheses.

- $LQ_{80} \to b\tau$ AUC = 0.93 (0.92[77]),
- $A_{50} \to 4\ell$ 0.93 (0.94[77]),
- $h_{60}^0 \to \tau\tau$ 0.85 (0.81[77]), and
- $h_{60}^{\pm} \to \tau\nu$ 0.94 (0.94[77]).

For the scenarios, the masses of the resonances are given in the subscript. Like the background, each signal scenario requires at least

**Table 1 | FPGA specifications and cost**

| | This paper | | This paper | | Govorkova et al.[77] | |
|---|---|---|---|---|---|---|
| **ML training and setup** | | | | | | |
| Framework | fwXmachina | | fwXmachina | | hls4ml | |
| Architecture | Deep decision tree | | Deep decision tree | | Variational autoencoder | |
| Dataset | γγjj | | LHC physics[83] | | LHC physics[83] | |
| Input variables | 8 | | 56 | | 56 | |
| No. of trees $T$ | 30 | | 30 | | NA for neural networks | |
| Max. depth $D$ | 6 | | 4 | | NA for neural networks | |
| Phys. performance | See text | | Comparable to[77] | | 77 | |
| **FPGA and firmware setup** | | | | | | |
| Chip family | Xilinx Virtex UltraScale+ | | Xilinx Virtex UltraScale+ | | Xilinx Virtex UltraScale+ | |
| Chip model | xcvu9p-flga2104-2L-e | | xcvu9p-flga2104-2L-e | | xcvu9p-flgb2104-2-e | |
| Platform | Vivado 2019.2 | | Vitis 2022.2 | | Vivado 2020.1 | |
| Clock | 200 MHz | 5 ns | 200 MHz | 5 ns | 200 MHz | 5 ns |
| Precision | ap_int⟨8⟩ | | ap_int⟨8⟩ | | ap_fixed⟨varies⟩ | |
| **FPGA cost** | | | | | | |
| Latency | 6 ticks | 30 ns | 6 ticks | 30 ns | 16 ticks | 80 ns |
| Interval | 1 tick | 5 ns | 1 tick | 5 ns | 1 tick | 5 ns |
| FF | 15 k | 0.6% | 15 k | 0.6% | 12 k* | 0.5% |
| LUT | 63 k | 5.4% | 109 k | 9.2% | 35 k* | 3% |
| DSP | 8 | 0.1% | 56 | 0.8% | 68* | 1% |
| BRAM | 0 | 0% | 0 | 0% | 13* | 0.3% |

The first column describes the design for γγjj; see text for details of the signal model on which the design is tested. The second column compares our result for the LHC physics problem given in the third column[77]. For the third column, the result listed is for DNN VAE PTQ 8-bit, the highlighted configuration in Ref. 77; the * indicates that the numbers are converted from the published percentages.

one electron or muon above the above-mentioned trigger threshold in the final state. The samples with $\tau$ lepton final states are dominated by the leptonic decays because of the trigger selection. Our AUC performance is comparable to the range of previous results[77].

For the FPGA cost, the configuration is run on an xcvu9p FPGA with a clock speed of 200 MHz. With similar physics performance compared to previous results[77], our FPGA resource utilization is at comparable values to the low end of the range of FF and LUT usage but fewer DSP and BRAM usage. Our design yields a lower latency value at six clock ticks (30 ns) and the lower bound of the range given at one clock tick (5 ns) for the interval. The results are summarized in the second column of Table 1.

As a cross-check of our FPGA cost, we implemented the two additional designs. The first cross-check uses only 26 variables on the same xcvu9p FPGA at 200 MHz. Due to the nature of the samples, many of the features are zero-valued, e.g., very few events have more than 3 jets. Therefore, we train with a subset of 26 input variables consisting of the $(p_T, \eta, \phi)$ for the 4 leading jets, 2 leading electrons, and 2 leading muons, along with $E_T^{miss}$ and its $\phi$ orientation. There is no difference in AUC using only 26 variables to within a percent of the 56 variable result above. The design is executed with a similar latency of seven ticks (35 ns) and the same interval of one tick (5 ns). However, the resource usage is significantly less than the 56 variable configuration at 9k FF, 61k LUT, 26 DSP, and no BRAM.

The second cross-check uses the 26 variable configuration on a smaller FPGA, on Xilinx Zynq UltraScale+ xczu7ev. The FPGA cost is nearly identical as reported above. The design is executed with the

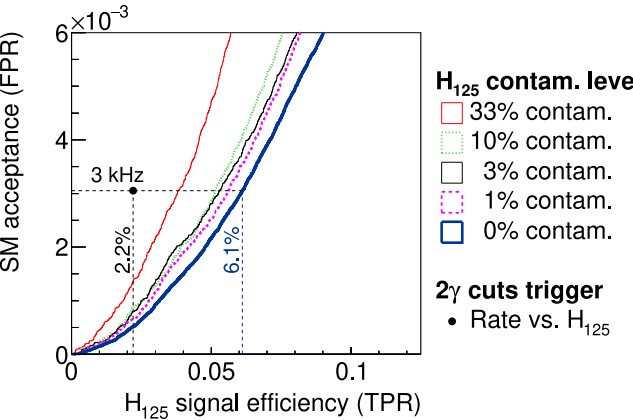

**Fig. 4 | ROC curves showing the SM$_{γγjj}$ acceptance vs. $H_{125}$ efficiency for different contaminated mixtures of $H_{125}$ that is used to train the autoencoder.** The legend indicates the percentage of the training sample consisting of $H_{125}$ with the rest consisting of the SM sample, i.e., the uncontaminated is trained only on the SM sample. The 3 kHz line and the values for the uncontaminated autoencoder trigger and the 2γ trigger matches that of the top-right plot in Fig. 3. The plot is software-simulated results using bit integers as done in the firmware.

same latency and interval; the resource usage is within 5% of the above values.

We note that the differences in the FPGA cost with respect to previous results[77] may be due to a number of factors. The factors include differences in the ML architecture as well as details about the FPGA configuration such as model compression methods, the number bits per input, type of input representation, such as fixed-point precision, and Xilinx versions.

With respect to the last item in the list, both Vivado HLS and Vitis HLS have been used to synthesize our designs with the latter being the more recent version of the same platform. Both are platforms that synthesize C code into an RTL implementation. For the benchmark scenario, the Vivado result is given in Table 1. The corresponding result using Vitis produced an increased latency value of 4 more ticks at the same clock speed, an increase of 50% increase in flip flops, and an increase of 30% in LUT with no change in DSP or BRAM. We have generally used Vivado to synthesize our designs, but it had difficulty with large designs such as the second configuration in Table 1. Although Vitis yielded a less performant FPGA design compared to Vivado for the benchmark, Vitis was able to synthesize the larger configuration for the comparison.

### Signal-contaminated training

A promising use case of the anomaly detector is to use collected data to train the autoencoder itself, rather than to use simulated samples, and to deploy it on subsequent incoming data. In this scenario, while the majority of the training sample would remain background, a fraction would consist of the signal since the data would contain the signal that would cause the anomaly. To study the autoencoder's performance using incoming data, we consider the results from the models trained with various levels of signal-contaminated simulated SM samples.

In Fig. 4, we show a family of ROC curves with varying levels of signal contamination in the training sample from 1% to a third of the total number of events. As expected, there is degradation of performance with an increasing fraction of the signal contamination in the training dataset. Nevertheless, training the autoencoder with a sample that has 33% contamination still outperforms the ATLAS-inspired diphoton trigger with about a factor of two higher $H_{125}$ acceptance at the same SM rate. Our findings are consistent with the anomaly detection study that reported a similar behavior for percent-level signal contamination[19].

For the benchmark physics process, an approximate upper bound of the signal contamination is estimated to be 1%. This bound considers known SM processes[94] and assumes that all Higgs bosons[102] decay to the $\gamma\gamma jj$ final state. Therefore, the resistance to contamination at the percent level—like that demonstrated in the study above—is promising for the rare BSM signals sought in high-energy physics experiments. A possible experimental setup to prepare for varying levels of contaminated data could be to employ a set of autoencoder triggers trained with varying levels of simulated signal contamination. A sketch of the setup is given in the Supplementary Fig. 4.

## Discussion

An implementation of a decision tree-based autoencoder anomaly detector was presented. The `fwXmachina` framework is used to implement the algorithm on FPGA with the goal of conducting real-time anomaly detection for physics beyond the Standard Model at real-time trigger systems at high-energy physics experiments. The implementation is tested on two problems: detection of exotic Higgs decays to $\gamma\gamma jj$ through pseudoscalar intermediates and an LHC physics anomaly detection dataset[83]. In both problems, the ML is trained only on background processes and evaluated on both signal and background. The anomaly detector shows the promise to identify several different realistic exotic signals that may be seen at a trigger system with comparable physics performance to existing neural network-based anomaly detectors. The efficient firmware implementation and low latency of 30 ns are well suited for the timing constraints of FPGA-based first-level triggers at LHC experiments.

A study of classifier performance with signal contamination shows the promise of the possibility of training on the collected data at the LHC. If the collected data already has BSM processes mixed in that we are trying to discover, then this possibility allows one to train the ML with the data anyway and then deploy it on future data to detect the BSM signal[103]. These approaches may also be of interest at the HL-LHC, which will increase the rate of proton collisions at the cost of higher background levels.

Existing approaches of the real-time trigger path anomaly detector, including the one in this paper, make assumptions about the availability of the preprocessed objects such as electrons that are reconstructed from more basic inputs such as calorimetric data. The next step would consider such inputs ranging from 1 k to 100 M channels, depending on the experimental setup, which may require a drastic redesign of existing approaches.

An added advantage of using decision tree-based anomaly detectors such as the algorithm presented here is that it allows for interpretability. As Fig. 1 and Supplementary Fig. 3 demonstrate, it is possible to examine the cuts used to construct the decision trees either by examining the feature space or the constructed trees. This enables visual interpretation of the anomaly detection. The large majority of autoencoders rely on neural networks and other black box models that have resisted easy interpretation[84] of the latent space and intermediate node values. Interpretability may be desirable in understanding trigger behavior in high-energy physics when disentangling BSM events from flaws in the apparatus leading to similar anomalous signals. Fields in which black box models are undesirable may also find our tool useful.

A challenging aspect of the analysis of anomalous events, which may affect other methods as well, is that the mapping of the input space to the anomaly score is not necessarily unique due to the Jacobian arising from the coordinate transformation[68]. That is, how rare a given event is depends on the choice of variables. In such cases, the events selected by a threshold on the score can be studied with variables orthogonal to the input space[74] or the latent space of the autoencoder[50]. Adding to the difficulty is what to do with the selected anomalous sample. We list three ideas in the literature that may help identify the BSM events in this sample. The first two methods use variables orthogonal to the input space. First, a bump hunt was conducted using invariant masses in Ref. 74. Second, a control sample could be obtained using a sideband to help identify the BSM events in the sample of anomalous events[48,49]. Lastly, an analysis of the latent space could help separate BSM from the other events[50]. For any of these methods, the BSM may not populate smoothly across the anomalous score distribution, so the BSM fraction would likely be extracted by a statistical treatment. As is commonly done in high energy physics, e.g., Ref. 104, a simultaneous maximum likelihood fit can extract the BSM composition in the various subsamples.

## Methods
### Details of simulated samples
Samples of the multistage process of simulating the proton collisions that produce our final state followed by the simulation of the detector effects, so called Monte Carlo samples, are considered in order to test the autoencoder's performance in real-time triggers.

We produced a sample of one million simulated proton-proton collision events in the SM composed of all processes that produce the $\gamma\gamma jj$ final state, which we consider the background process during the evaluation of physics performance.

Additionally, two signal samples of one hundred thousand events each that simulate the production and decay of scalar bosons are generated, which we consider the anomaly processes. Scalar bosons produced from the gluon-gluon fusion production mode in proton-proton collisions are decayed as $H_{125} \to a_{10}a_{70}$ and $H_{70} \to a_5 a_{50}$. The lighter $a$ decays to $\gamma\gamma$ and the heavier $a'$ decays to $jj$. All samples, both background and anomaly, use the Higgs effective field theory model in MadGraph5_aMC 2.9.5[94].

The input variables are the reconstructed values calculated by Delphes 3.5.0[96,97]. Jets are reconstructed with the anti-$k_t$ algorithm with a radius parameter $R = 0.4$ and a minimum $p_T$ of 20 GeV[105]. Photons are reconstructed with a radius parameter of $R = 0.2$ and a minimum $p_T$ of 0.5 GeV. All samples are produced with the above-mentioned Mad-Graph5 and decayed and showered with Pythia8[95]. Detector simulation and event reconstruction are simulated with Delphes, which uses the CMS card to simulate the behavior of the CMS detector[98]. We note the similarities between the physics capabilities of the CMS and ATLAS detectors allow a generic interpretation of the results presented in the next section. Without mitigation, multiple proton-proton interactions (pileup) impact the number of jets reconstructed in each event. Due to the importance of hadronic jets in the HL-LHC, a variety of algorithms have been proposed for removing pileup contributions in jets[106–108], and therefore we neglect the effects of pileup. More details can be found in the samples[109]. The input variable distributions are given in Fig. 2.

### Firmware design
The structure of the firmware is based on `fwXmachina`[78,79]. The AUTOENCODER PROCESSOR, whose block diagram is shown in Fig. 5, takes in input data and outputs the anomaly score. In the firmware implementation, we approximate $\mathbb{R}$ of the input-output space by $N$-bit integers $\mathbb{Z}_N$.

In the diagram, input enters from the left, and copies are distributed to $T$ deep decision trees, each tree corresponding to one latent dimension. Once the outputs of the engine are available, the distance processor computes the $\Delta$ with respect to the input. The DEEP DECISION TREE ENGINE (DDTE)[79] is modified to output a vector of values. The DISTANCE PROCESSOR takes the outputs of DDTE and computes the distance for each set of outputs followed by a sum.

We note that further modification of DDTE would allow for efficient transmission of compressed data[110], but is beyond the scope of this paper.

### Verification and validation
We validate and verify our design using the benchmark physics scenario.

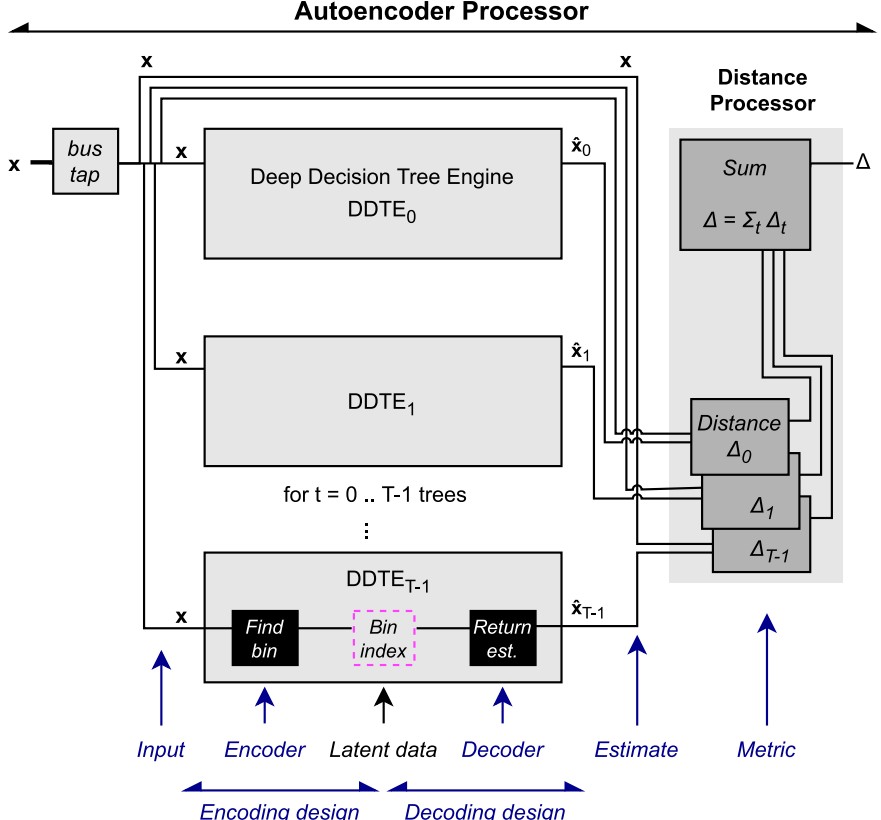

**Fig. 5 | Block diagram of the AUTOENCODER PROCESSOR for anomaly detection with a $T$-dimensional latent layer corresponding to a forest of $T$ decision trees.** The design uses DEEP DECISION TREE ENGINE [79], as both the encoder and decoder with the bin index are shown only schematically, as the latent data is implicit.

For validation of our algorithm, first we run $\mathcal{O}(10^5)$ test vectors through our design using C simulation in Vivado HLS and compare the outputs to that of the expected firmware outputs simulated in Python. Then co-simulation is done, which creates an RTL model of the design, simulates it, and compares the RTL model against the C design. In all cases, the simulation outputs match the expected outputs.

For the physical verification of our algorithm, we program select configurations onto the xcvu9p at a clock speed of 200 MHz, which is the setup used for the benchmark results in this paper. We test a handful of test vector inputs and use the Xilinx Integrated Logic Analyzer IP core to observe the outputs. In all cases, the outputs match the expected outputs received from software and co-simulation.

## Data availability

Two datasets were used in this paper. The $\gamma\gamma jj$ data generated by us for this study have been deposited in Mendeley Datasets under https://doi.org/10.17632/44t976dyrj.1 and is cited as Ref. 109. The LHC physics dataset was taken from Ref. 83 and is publicly available in Zenodo under https://doi.org/10.5281/zenodo.3675210, https://doi.org/10.5281/zenodo.3675206, https://doi.org/10.5281/zenodo.3675203, https://doi.org/10.5281/zenodo.3675199, and https://doi.org/10.5281/zenodo.5046388.

## Code availability

The repository with the files to evaluate the FPGA performance is publicly available at D-Scholarship@Pitt, which is an institutional repository for the research output of the University of Pittsburgh[111]. More specifically, the IP core design for the benchmark scenario is available, along with a testbench and associated test vectors. General information about `fwXmachina` can be found at http://fwx.pitt.edu.

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

## Acknowledgements

We thank David Shih, Matthew Low, Joseph Boudreau, James Mueller, Elliot Lipeles, Dylan Rankin, and Eli Ullman-Kissel for the physics discussions. We thank Kushal Parekh, Stephen Racz, Brandon Eubanks, Yuvaraj Elangovan, and Kemal Emre Ercikti for the firmware discussions. We thank Santiago Cané for assistance in testing. We thank Gracie Jane Gollinger for computing infrastructure support. TMH was supported by the US Department of Energy [award no. DE-SC0007914]. JS was supported by the US Department of Energy [award no. DE-SC0012704]. BTC was supported by the National Science Foundation [award no. NSF-2209370]. STR was supported by the Emil Sanielevici Undergraduate Research Scholarship.

## Author contributions

S.T.R., J..S, W.C.O., and T.M.H. designed the ML training algorithm. Q.B. and P.S. implemented and tested the firmware design. S.T.R., B.T.C., and T.M.H. created the simulated dataset and performed the physics analysis. S.T.R. led the project execution, while T.M.H. managed and coordinated the overall effort. T.M.H. and S.T.R. drafted and edited the manuscript with significant input from BTC. All authors reviewed the manuscript.

## Competing interests

The authors declare competing interests. TMH, BTC, JS, STR, and QB have filed a patent on the firmware design of the autoencoder with the University of Pittsburgh. It is currently pending as US Patent Application Publication No. US 2024/0054399. Other authors declare no competing interests.
