## [Peer Review File · Nature Communications]

Nanosecond anomaly detection with decision trees and real-time application to exotic Higgs decaysReviewer #1 (Remarks to the Author):

- What are the noteworthy results?

The noteworthy result here is a machine-learning (autoencoder) approach to identify anomalies in high energy particle collisions in an extremely fast (25 ns) and low resource implementation (in FPGAs) to enable experiments to select such event in the first level of their "trigger" systems.

- Will the work be of significance to the field and related fields? How does it compare to the established literature? If the work is not original, please provide relevant references.

The work will be significant in the field of collider particle physics as it demonstrates that enabling a new way of identifying new features in data using an autoencoder is very achievable using modern FPGAs. It could apply to other fields that have high bandwidth data streams and desire a way to find anomalies in real-time, whatever the source, that stick out compared the "normal" data.

This work builds on earlier approaches, and exceeds those by being more compact in FPGA resource usage and being faster. In other words, this is not the first use of an autoencoder in our field, nor of machine-learning in FPGAs, but the authors are achieving superior results. The authors do explore using a forest of decision trees rather than a neural network to implement the anomaly detection. The authors also cite earlier work on the same or related topics.

- Does the work support the conclusions and claims, or is additional evidence needed?

The work generally supports the conclusions, aside from a remark below on whether the achieved anomaly performance is really applicable to the trigger level with presumably coarser resolutions, and two other minor remarks. Regarding the latter minor points, the claims made in the abstract (and elsewhere) about percent level (FPGA) resource usage needs some point of reference (e.g. the specific FPGA family and/or size). And while the authors note that they achieved inferences with a 25 ns latency, which make it applicable to the trigger level of an LHC experiment, this is not a strict constraint. As long as the trigger logic (inference) is pipelined at 25 ns, it also can run in the trigger level of an LHC experiment (as is done today). But it is true that 25 ns would be a very fast inference indeed.

- Are there any flaws in the data analysis, interpretation and conclusions? - Do these prohibit publication or require revision?

The authors suggest that their implementation would enable selecting anomalous events in the first "trigger" levels of a collider experiment, and give several benchmark scenarios to demonstrate that. However, their simulated data samples make use of the Delphes detector performance simulation, which to my knowledge would generally assume detector reconstruction performance and resolution appropriate for a complete offline data analysis, not for the first trigger level where measurements are coarser and more limited. At the trigger level, I would expect the resolution to measure jets, muons, and electrons would be worse than that available offline (which uses precision tracking detectors, for example). Therefore, the momentum and mass plots shown in Fig.3 would probably exhibit much less sharp peaks, and that would likely degrade the anomaly score ultimately assigned to events. i.e. the simulated Higgs boson mass peaks would be much broader. So while I do see value in such an approach of sifting through data for anomalies, its application to an experiment's trigger level needs some clarification here on the resolutions assumed and possibly would require reassessing the ROC curve performance if those resolutions should be degraded to match what is appropriate at the trigger level.

On a minor point, the different FPGA resource usage results in Table 1 could also be susceptible the version of the FPGA synthesis compiler used (namely Vivado vs. Vitis), as my group has experienced,

so it might be mentioned at the end of section 3.2 as another source of possible uncertainty when comparing different results across the table.

- Is the methodology sound? Does the work meet the expected standards in your field?

Analysis of the details of the machine-learning methodology employed (Section 2) requires an expert more familiar than me to judge its soundness. However I have no reason to doubt that the authors know what they are doing, and the usage of autoencoders for anomaly detection in the field of particle physics is well accepted.

The methodology used to obtain physics performance scores (i.e. ROC curves) needs at a minimum some clarification of what resolutions went into the simulation and their applicability to the trigger, or possibly a reassessment using resolutions expected for the claimed trigger application.

Otherwise the authors seem to have done a good job developing, training, implementing, and measuring the performance of an FPGA-based autoencoder for the purposes of anomaly detection. To me these aspects meet the standards of the field.

- Is there enough detail provided in the methods for the work to be reproduced?

Yes, generally there is enough information aside from the remark above about the details of the simulation. However there is one other point that needs clarification. Section 2.3 writes that the input variables to the autoencoder are the three invariant masses ee , $mumu$, and $eemumu$. But the caption to Fig.3 says the input variable distributions are leading lepton PT, two lepton mass, and four-lepton mass. So, is leading lepton PT one of the input variables or not? Needs clarification.

Reviewer #2 (Remarks to the Author):

This article presents a DDT (Deep Decision Tree) implementation on an FPGA based on the FwXmachina framework. The method is applied to the search of anomalies using LHC simulated datasets (the standard model being viewed as a background). To my knowledge, previous works are based on neural network and as such this work is original. The performances are equivalent with previous NN reported results but the approach is different leading to a lower latency. To support their claim, the authors use two data sets, one from a previous study by Govorkova et al. and one that they generated themselves. The author provides the dataset and the FPGA frameworks on a dedicated website to be able to reproduce their results.

This work follows two articles on their works on the FwXmachina framework:

- 1) Classification with flat tree architecture_ in the Journal of Instrumentation [JINST 16 P08016 (2021)](<https://dx.doi.org/10.1088/1748-0221/16/08/P08016>)
- 2) Regression with deep end-to-end decision tree architecture_ in the Journal of Instrumentation [JINST 17 P09039 (2022)](<https://doi.org/10.1088/1748-0221/17/09/P09039>)

The main difference is the introduction of a training on background only method in order to be able to detect anomalous events (Their previous work used a BDT trained with the root TMVA package).

I have no expertise in Higgs physics and in the ATLAS/CMS experiment and I cannot assess this part.

I have the following remarks :

- 1) The inferences in 25 ns correspond to a very specific case ($e+e-u+u-$ dataset with very few input variables). In the other presented implementation of their work, the latency is higher (30 ns and 35 ns). Therefore, the statement "we present a novel implementation ... that make inferences in 25 ns"

looks incorrect and too strong for me.

3) On page 8, it is reported that the events are simulated using Delphes the CMS detector while the authors propose an application to the ATLAS L1/L0 case. Why the authors did not simulate the behavior of the ATLAS detector in Delphes ? It do not change the conclusions but deserve a few words of explanation.

4) The authors proposes to use a dedicated processor near the FPGA using trigger data and a multi-event analysis to implement a window selection on the anomaly score. It looks like a complex object. Can they elaborate on its implementation in the constrained environnement of a trigger path ?

5) The authors compare their work to "Autoencoders on FPGAs for real-time, unsupervised new physics detection at 40 MHz at the Large Hadron Collider" (Govorkava et al.). It will be more useful and help the readability of their paper if they limit the comparison with the architecture eventually selected in the Govorkava paper (DNN VAE PTQ 8 bits). Using a range is very misleading.

6) The DDN implemented for the two dataset use two different tools (Vivado 2019.2 and Vitis 2022.2). Given the time needed for a compilation (<1h). Does Vitis gives the same results on their first design ?

Reviewer #3 (Remarks to the Author):

This paper reports a decision-tree-based autoencoder, which can run inference very fast (comparable to the LHC lowest level trigger latency) and has been demonstrated with a number of simulated datasets. This is a serious study, the topic is timely, and the manuscript is well written.

I am generally not convinced by the technical solution or by the applications and so have difficulty recommending publication of the current paper. Here is the longer version:

Technical solution. As the authors note, there have been many papers proposing autoencoders (online) anomaly detection, but I don't yet see how a vanilla autoencoder could be used for this purpose in practice. For example, (1) autoencoders are not targeting new physics, they are targeting "rare" events so there is an assumption that new = rare (fine, but should be stated explicitly). Rare is also not even a well-defined quantity as it depends on the coordinates. (2) a vanilla autoencoder (like the one proposed here) is not even learning rarity (=density), there is a reliance on information bottlenecks (i.e. uncontrolled ineffectiveness). This would at least be solved with the equivalent of a VAE. (3) What do you do with the selected events? It seems like you are assuming 100% knowledge of the background, but this is far from reality. All of this is related to the fact that you do not compare the physics performance with any other method (hard to compare methods when the target is not clear).

[Why not just use a NN VAE and then train a DT to emulate that?]

In my opinion, the latency challenge is not the most pressing issue for online anomaly detection. On the other hand, the latency of the algorithm presented in this paper is impressive!

Applications: Do we really need anomaly detection for final states with 4 leptons? I really don't think there is a qualitative gain from this path. I appreciate that the authors stress that they focus on events that would fail conventional trigger paths, but I did not see the actual benefits? e.g. the cross sections sensitivity improves by X%. My guess is that X will not be very impressive.

My concerns also apply to [69] so I will defer to the editors on how to best consider them (or not).

Reviewer #1 (Remarks to the Author):

- What are the noteworthy results?

The noteworthy result here is a machine-learning (autoencoder) approach to identify anomalies in high energy particle collisions in an extremely fast (25 ns) and low resource implementation (in FPGAs) to enable experiments to select such event in the first level of their “trigger” systems.

- Will the work be of significance to the field and related fields? How does it compare to the established literature? If the work is not original, please provide relevant references.

The work will be significant in the field of collider particle physics as it demonstrates that enabling a new way of identifying new features in data using an autoencoder is very achievable using modern FPGAs. It could apply to other fields that have high bandwidth data streams and desire a way to find anomalies in real-time, whatever the source, that stick out compared the “normal” data.

This work builds on earlier approaches, and exceeds those by being more compact in FPGA resource usage and being faster. In other words, this is not the first use of an autoencoder in our field, nor of machine-learning in FPGAs, but the authors are achieving superior results. The authors do explore using a forest of decision trees rather than a neural network to implement the anomaly detection. The authors also cite earlier work on the same or related topics.

- Does the work support the conclusions and claims, or is additional evidence needed?

The work generally supports the conclusions, aside from a remark below on whether the achieved anomaly performance is really applicable to the trigger level with presumably coarser resolutions, and two other minor remarks. Regarding the latter minor points, (a) the claims made in the abstract (and elsewhere) about percent level (FPGA) resource usage needs some point of reference (e.g. the specific FPGA family and/or size). (b) And while the authors note that they achieved inferences with a 25 ns latency, which make it applicable to the trigger level of an LHC experiment, this is not a strict constraint. As long as the trigger logic (inference) is pipelined at 25 ns, it also can run in the trigger level of an LHC experiment (as is done today). But it is true that 25 ns would be a very fast inference indeed.

** Thank you for this observation. We agree and add appropriate text whenever relevant. For example, we added “Xilinx Virtex UltraScale+ VU9P FPGA” in the abstract. We agree about the point regarding the 25 ns if standalone, but we’d like to point out that there are constraints in integrated systems. For HL-LHC ATLAS, level-0 Global Trigger algorithms are limited to ~ 100 ns, but this is beyond the scope of the paper.

- Are there any flaws in the data analysis, interpretation and conclusions? - Do these prohibit publication or require revision?

The authors suggest that their implementation would enable selecting anomalous events in the first “trigger” levels of a collider experiment, and give several benchmark scenarios to demonstrate that. However, their simulated data samples make use of the Delphes detector performance simulation, which to my knowledge would generally assume detector reconstruction performance and resolution appropriate for a complete offline data analysis, not for the first trigger level where measurements are coarser and more limited. At the trigger level, I would expect the resolution to measure jets, muons, and electrons would be worse than that available offline (which uses precision tracking detectors, for example). Therefore, the momentum and mass plots shown in Fig.3 would probably exhibit much less sharp peaks, and that would likely degrade the anomaly score ultimately assigned to events. i.e. the simulated Higgs boson mass peaks would be much broader. So while I do see value in such an approach of sifting through data for anomalies, its application to an experiment’s trigger level needs some clarification here on the resolutions assumed and possibly would require reassessing the ROC curve performance if those resolutions should be degraded to match what is appropriate at the trigger level.

** Thank you for the comment. We did use offline values and agree that trigger level resolutions should be used for the performance evaluation. However, there are many upgrades for HL-LHC that close the gap between the trigger and offline. For instance, the FPGA-based trigger system of the ATLAS experiment will receive the full EM calorimeter cells similar to offline, so photon performance would be fairly close. As for jets, the ATLAS experiment plans to form offline-like anti-kt jets in the FPGA, so the jet performance would be fairly close. We added more details in the text and Appendix A.

On a minor point, the different FPGA resource usage results in Table 1 could also be susceptible the version of the FPGA synthesis compiler used (namely Vivado vs. Vitis), as my group has experienced, so it might be mentioned at the end of section 3.2 as another source of possible uncertainty when comparing different results across the table.

** Thank you for the comment. We had trouble running Vivado on the dataset of Govorkova et al., which is why we switched to Vitis. So we ran the new benchmark point using Vivado and Vitis at the same clock speed. Contrary to what one might have expected, we observe a less performant design using Vitis compared to the one from Vivado. Therefore, we do not suspect that Vitis is giving a benefit from a performance perspective, but rather it allows for the synthesis in a more robust way. We note our observations in a new paragraph at the end of Sec. 3.2.

- Is the methodology sound? Does the work meet the expected standards in your field?

Analysis of the details of the machine-learning methodology employed (Section 2) requires an expert more familiar than me to judge its soundness. However I have no reason to doubt that the authors know what they are doing, and the usage of autoencoders for anomaly detection in the field of particle physics is well accepted.

The methodology used to obtain physics performance scores (i.e. ROC curves) needs at a minimum some clarification of what resolutions went into the simulation and their applicability to the trigger, or possibly a reassessment using resolutions expected for the claimed trigger application.

** We use offline object quantities converted to bitwise input variables. With HL-LHC upgrades we believe that offline-like variables will be available at level-1, so that would be a reasonable approximation. For the FPGA performance, we convert these values to bitwise quantities for evaluation. We have added some sentences in the text to clarify this procedure.

Otherwise the authors seem to have done a good job developing, training, implementing, and measuring the performance of an FPGA-based autoencoder for the purposes of anomaly detection. To me these aspects meet the standards of the field.

- Is there enough detail provided in the methods for the work to be reproduced?

Yes, generally there is enough information aside from the remark above about the details of the simulation. However there is one other point that needs clarification. Section 2.3 writes that the input variables to the autoencoder are the three invariant masses ee , $mumu$, and $eemumu$. But the caption to Fig.3 says the input variable distributions are leading lepton p_T , two lepton mass, and four-lepton mass. So, is leading lepton p_T one of the input variables or not? Needs clarification.

** In our new draft, we replaced the benchmark and use different input variables. So this particular comment does not apply anymore. Nevertheless, we clarify the situation for the previous draft. The leading lepton p_T was used to exclude events with a leading lepton $p_T > 23$ GeV—these are the events that would already be triggered by the conventional lepton trigger—but the variable itself was not included in the training. The previous training involved only the three invariant masses.

Reviewer #2 (Remarks to the Author):

This article presents a DDT (Deep Decision Tree) implementation on an FPGA based on the FwXmachina framework. The method is applied to the search of anomalies using LHC simulated datasets (the standard model being viewed as a background). To my knowledge, previous works are based on neural network and as such this work is original. The performances are equivalent with previous NN reported results but the approach is different leading to a lower latency.

To support their claim, the authors use two data sets, one from a previous study by Govorkava et al. and one that they generated themselves. The authors provide the dataset and the FPGA frameworks on a dedicated website to be able to reproduce their results.

This work follows two articles on their works on the FwXmachina framework:

1) Classification with flat tree architecture_ in the Journal of Instrumentation [JINST 16 P08016 (2021)](<https://dx.doi.org/10.1088/1748-0221/16/08/P08016>)

2) Regression with deep end-to-end decision tree architecture_ in the Journal of Instrumentation [JINST 17 P09039 (2022)](<https://doi.org/10.1088/1748-0221/17/09/P09039>)

The main difference is the introduction of a training on background only method in order to be able to detect anomalous events (Their previous work used a BDT trained with the root TMVA package).

I have no expertise in Higgs physics and in the ATLAS/CMS experiment and I cannot assess this part.

I have the following remarks :

1) The inferences in 25 ns correspond to a very specific case (e+e-u+u- dataset with very few input variables). In the other presented implementation of their work, the latency is higher (30 ns and 35 ns). Therefore, the statement "we present a novel implementation ... that make inferences in 25 ns" looks incorrect and too strong for me.

**** We changed "25 ns" to "30 ns" to match our new result.**

3) On page 8, it is reported that the events are simulated using Delphes the CMS detector while the authors propose an application to the ATLAS L1/L0 case. Why the authors did not simulate the behavior of the ATLAS detector in Delphes ? It does not change the conclusions but deserves a few words of explanation.

** Only mentioning ATLAS was an oversight and we revised to text to mention that the approach may be taken by ATLAS or CMS.

4) The authors proposes to use a dedicated processor near the FPGA using trigger data and a multi-event analysis to implement a window selection on the anomaly score. It looks like a complex object. Can they elaborate on its implementation in the constrained environment of a trigger path ?

** The algorithm like the one proposed in our paper would be implemented in, for example, the Global Trigger system of the Phase-2 Upgrade of the TDAQ system for the ATLAS experiment (see, Sec. 9 starting on p125 on <https://cds.cern.ch/record/2285584/files/ATLAS-TDR-029.pdf>). They present a system using Xilinx VU13P, which is similar to the one used in our paper (Xilinx VU9P). Such a system is capable of inputs such as invariant mass, DeltaR, and pT for the DT-AE. Although Figure 9.1 in the document above does not show too much detail, the DT-AE could sit in the Global Event Processor where the block that replaces the L1Topo functionality (<https://iopscience.iop.org/article/10.1088/1742-6596/898/3/032037/pdf>). With respect to the timing constraint (Table 9.5) and the resource constraint (Sec. 9.6.2), we believe that an efficient DT-AE can be implemented in such a system. This is a bit too much detail for our paper, but we have added references to the Phase-2 upgrade of TDAQ for ATLAS and CMS.

5) The authors compare their work to "Autoencoders on FPGAs for real-time, unsupervised new physics detection at 40 MHz at the Large Hadron Collider" (Govorkava et al.). It will be more useful and help the readability of their paper if they limit the comparison with the architecture eventually selected in the Govorkava paper (DNN VAE PTQ 8 bits). Using a range is very misleading.

** Changed to only list the "DNN VAE PTQ 8 bit" results in the table and text.

6) The DDN implemented for the two dataset use two different tools (Vivado 2019.2 and Vitis 2022.2). Given the time needed for a compilation (<1h). Does Vitis gives the same results on their first design ?

** Thank you for the comment. We had trouble running Vivado on the dataset of Govorkova et al., which is why we switched to Vitis. So we ran the new benchmark point using Vivado and Vitis at the same clock speed. Contrary to what one might have expected, we observe a less performant design using Vitis compared to the one from Vivado. Therefore, we do not suspect that Vitis is giving a benefit from a performance perspective, but rather it allows for the synthesis in a more robust way. We note our observations in a new paragraph at the end of Sec. 3.2.

Reviewer #3 (Remarks to the Author):

This paper reports a decision-tree-based autoencoder, which can run inference very fast (comparable to the LHC lowest level trigger latency) and has been demonstrated with a number of simulated datasets. This is a serious study, the topic is timely, and the manuscript is well written.

I am generally not convinced by the technical solution or by the applications and so have difficulty recommending publication of the current paper.

**** Thank you for your comments. To address many of your points, we conducted a new study to replace our benchmark. Detailed in-line responses are given below.**

Here is the longer version:

Technical solution.

As the authors note, there have been many papers proposing autoencoders (online) anomaly detection, but I don't yet see how a vanilla autoencoder could be used for this purpose in practice.

For example,

- (1) autoencoders are not targeting new physics, they are targeting "rare" events so there is an assumption that new = rare (fine, but should be stated explicitly).

**** Thank you for the comment. We agree and made changes in the text to clarify this point. In many instances, we add "rare" to accompany BSM. We replace "detecting" / "identify" with "triggering on".**

Rare is also not even a well-defined quantity as it depends on the coordinates.

**** We're not quite sure, but we think you mean the mathematical property of the trigger algorithm. Let's say that x represents the anomaly score and the corresponding event rate as $f(x)$. We can define rarity by choosing a value y that is acceptable to the experiment. For example, we choose $y = 3$ kHz that corresponds to the diphoton trigger rate at the ATLAS experiment in 2018. With this requirement of $y = f(x_{\text{threshold}})$, we identify $x_{\text{threshold}}$ for the anomaly score. Since $f(x)$ is a monotonically increasing function, we believe that it is well-defined.**

The metric in use is a metric in the mathematical sense, and thus isometric (respects coordinate transformation). But as noted in our response, the meaning of "distance" with respect to the different variables is changed by normalization (and is in that sense

arbitrary). However, it didn't have physical significance to begin with. We don't see a way around this.

Normalization does preserve "rarity" in the sense that the "rare events" that we identify are more unusual than nominal events in both absolute and relative terms *per coordinate*. All normalization does is put each coordinate on equal footing, which we think is a reasonable assumption if one doesn't know anything about the data a priori.

- (2) vanilla autoencoder (like the one proposed here) is not even learning rarity (=density), there is a reliance on information bottlenecks (i.e. uncontrolled ineffectiveness). This would at least be solved with the equivalent of a VAE.

** Thank you for the comment. Our responses to your two points are below.

Density) The construction of our DT-AE is directly based on density estimation. The relevant step in the construction is the sampling of the PDF of the training sample to determine cut to split of the sample. This is Steps 6 and 8 of the pseudocode, corresponding to choosing which variable to cut on followed by a cut on the chosen variable, respectively. More visually, the first two steps of Figure 1 show the 1d projections where the splits are made. Lastly, Figure A.3 shows how the bin sizes get smaller near areas of high density as the tree depth is increased. This is evidence that our DT-AE is directly learning the density of the data sample.

Information bottleneck) We comment on the dimensionality of the latent space and its relevance to the information bottleneck. The dimensionality of the latent space of our DT-AE is the number of trees used. This is because each DT produces the bin number, which corresponds to the encoded data, which is subsequently decoded for the estimate. Our implementation bypasses this encoding-and-decoding step for convenience, but it is physically producible if one desires to access that information. And so the dimensionality for our setup can easily exceed the dimensionality of the input variables, and it does so without any degradation of physics performance, although it comes at the price of firmware implementation. Therefore, our DT-AE does not rely on an information bottleneck since the dimensionality can easily exceed the input space. We added a paragraph at the end of Sec. 2.2 about this.

- (3) What do you do with the selected events?

** In our paper we are focused on the FPGA-based trigger system, so we do not process the events further. In an experiment such as ATLAS or CMS, further refinement would be done using a farm of CPU-based high level trigger (HLT) system. After the events are saved to disk, full reconstruction of the raw data would be done and a detailed analysis would be conducted. However, all of this is beyond the scope of the paper.

It seems like you are assuming 100% knowledge of the background, but this is far from reality.

** We agree with the comment, but our assumption does not conflict with it either. With our simulated sample we show the effectiveness assuming 100% knowledge. But more than that, we show in Sec. 3.3 the performance when the training sample is contaminated with the signal itself, which would be the worst kind of contamination. Lastly, we note that deployment itself can use real data taken by the experiments to train and a possible configuration in a diagram in the appendix, now Fig. B.4.

All of this is related to the fact that you do not compare the physics performance with any other method (hard to compare methods when the target is not clear).

** We do two comparisons. This comment actually convinced us that we need a better example to showcase our intent, so conducted the new study for the new benchmark. (1) Our new benchmark compares DT-AE with the existing diphoton trigger. (2) This part is already in the earlier draft: We compare the DT-AE tool with that of the NN-AE tool from [69] (Govorkova et al.) using their dataset available online. Please see the bottom row of Fig. 4 and the right two columns of Table 1.

[Why not just use a NN VAE and then train a DT to emulate that?]

** This is an interesting idea that is certainly worth investigation. We note that our DT-AE already achieves nearly identical performance as the VAE when using VAE tool on their dataset. DT-AE emulating a VAE would likely perform worse than the VAE itself since it would be an approximation.

In my opinion, the latency challenge is not the most pressing issue for online anomaly detection. On the other hand, the latency of the algorithm presented in this paper is impressive!

** Latency is one of the major challenges. Take, for instance, ATLAS Phase-2 Upgrade (<https://cds.cern.ch/record/2285584/files/ATLAS-TDR-029.pdf>). Table 5.5 on p. 102 shows that the algorithm latency for Global Trigger pipelined processing is at 100 ns. The details are currently being worked out, but this sets the scale of the timing required.

Applications:

Do we really need anomaly detection for final states with 4 leptons? I really don't think there is a qualitative gain from this path. I appreciate that the authors stress that they focus on events that would fail conventional trigger paths, but I did not see the actual benefits? e.g. the cross sections sensitivity improves by X%. My guess is that X will not be very impressive.

My concerns also apply to [69] so I will defer to the editors on how to best consider them (or not).

** Thank you very much for this comment. This point made us realize that we need to replace the benchmark with a new study that demonstrates a bottleneck due to current approaches. After conducting a few prototype studies, we settled on a physics scenario that is related to the one we already presented, but with a part of the phase space that remains unexplored using current approaches. Instead of $H \rightarrow aa \rightarrow ee \mu\mu$, we now present $H \rightarrow aa \rightarrow \gamma\gamma jj$, where j represents hadronic jets. The latest experimental result from ATLAS uses a diphoton trigger (requiring $\gamma\gamma$ above a E_T threshold of about 25 GeV), and so they do not explore the region of $m_a < 20$ GeV. We believe that the improved photon triggers at the HL-LHC can bring some sensitivity in this region with offline-like information available at level-1 triggers, and we compare its performance with that of the DT-AE on this region. We find the DT-AE brings significant improvement in sensitivity.

Reviewer #1 (Remarks to the Author):

- What are the noteworthy results?

The primary message of this revised manuscript remains to report on a machine-learning (autoencoder) approach to improve the sensitivity to new physics through anomalies in high energy particle collision data using an extremely fast (30 ns) and low resource implementation (in FPGAs) to enable experiments to select such event in the first level of their real-time data selection ("trigger") systems. The have increased the significance of their approach by adding a new study of a more challenging physics channel.

- Will the work be of significance to the field and related fields? How does it compare to the established literature? If the work is not original, please provide relevant references.

The work is significant in the field of collider particle physics as it demonstrates that enabling a new way of identifying new features in data using an autoencoder is very achievable using modern FPGAs and can fit the tight timing constraints of collider trigger systems. The authors provide an extensive list of references of usage of neural networks in general to identify new features in data. The results here, which apply to the more constrained environment of online selection, apply to other fields that have high bandwidth data streams and a need to identify anomalies in real-time.

The authors achieve a very compact and fast (30ns) implementation of an autoencoder able to identify a new physics channel (Higgs decay to pseudoscalars, in a different and more challenging final state than in the initial version of this article) with more sensitivity than traditional trigger methods. They also compare an implementation to a previously published implementation that used a different architecture, and for the same FPGA device, same clocking, and same physics samples achieve a faster implementation for similar resource usage.

- Does the work support the conclusions and claims, or is additional evidence needed?

The reported work does support the conclusions. The authors addressed an earlier referee comment questioning the usage of offline precision quantities for their autoencoder to demonstrate sensitivity, and (successfully) make the case that for the HL LHC upgrades of the CMS and ATLAS experiments the precision indeed would be at about that level in their trigger systems.

- Are there any flaws in the data analysis, interpretation and conclusions? - Do these prohibit publication or require revision?

The authors have cleared up some previous reservations about the first version of the article. Aside from their argument justifying use of offline precision variables, they also standardized on a specific FPGA device and clocking frequency to make comparisons across physics models and to previous work easier. Additionally, the authors have made their use-case even stronger by studying a more challenging new physics decay channel to demonstrate that their anomaly-driven approach can achieve factors of 3 to 100 times more sensitivity than traditional cut-based trigger approaches.

- Is the methodology sound? Does the work meet the expected standards in your field?

The work appears sound and is at the expected level of standards for our field.

- Is there enough detail provided in the methods for the work to be reproduced?

Yes, I believe there is enough detail here for reproducibility.

Reviewer #1 (Remarks on code availability):

Actually the link for the code repository requires a login account.

Reviewer #2 (Remarks to the Author):

The authors have take into account my concerns. The manuscript has greatly improved.

Reviewer #2 (Remarks on code availability):

1) The provided link (<http://d-scholarship.pitt.edu/id/eprint/45784>) for accessing the code requires a login and a password that I do not have.

2) Accessing the code through <https://www.fwx.pitt.edu> allows to install the fwMachina framework. The code is documented using Doxygen (with a few minor bugs in the rendering.)

The readme provides all the installation step. It is well explained but the provided method does not work on lxplus (as advertised) but work at ccin2p3. The "hello world" example is working fine. I was not able to find easily how to reproduce the article result (maybe because of 1). A readme in the article folder would have been helpful.

3) I do not have the Xinlinx tools installed so I am not able to test this part

Reviewer #3 (Remarks to the Author):

Thank you for taking into account my feedback! I particularly appreciate the new benchmark study. I have two follow up points:

1. Sorry for not being clear about "coordinate dependence". By this I mean if you have your features x and a score that is related to $p(x)$, so you keep events with $p(x) < \text{cut}$, then if $x \rightarrow f(x)$, the events you select need not be the same (even for the same fraction of events selected). This is because there is a Jacobian factor that $p(x)$ picks up when going from x to $f(x)$ and this may not preserve the order of events. This is not a problem per se, but it does limit the interpretability and universality of unsupervised anomaly detection (it is not a challenge just for this paper, but maybe useful to state).

2. [more importantly] I agree that latency in general is a problem of the HL-LHC, but I remain unconvinced that it is the main problem for anomaly detection. The crux is this: "After the events are saved to disk, full reconstruction of the raw data would be done and a detailed analysis would be conducted. However, all of this is beyond the scope of the paper." What, exactly, do you propose you do with the anomalous events that you save? I am not sure what analysis you will conduct. You need to know how anomalous the events are, which your method does not tell you. One answer to this might be "as a baseline, we compare to simulations". I don't find this very satisfying (after all, we are likely operating in a region where the simulations are not accurate), but at least it would supply one possible use of the data. For offline analyses, background estimation is a critical component. I appreciate that this criticism also applies to Govorkova et al.

Reviewer #1 (Remarks to the Author):

- What are the noteworthy results?

The primary message of this revised manuscript remains to report on a machine-learning (autoencoder) approach to improve the sensitivity to new physics through anomalies in high energy particle collision data using an extremely fast (30 ns) and low resource implementation (in FPGAs) to enable experiments to select such event in the first level of their real-time data selection (“trigger”) systems. The have increased the significance of their approach by adding a new study of a more challenging physics channel.

- Will the work be of significance to the field and related fields? How does it compare to the established literature? If the work is not original, please provide relevant references.

The work is significant in the field of collider particle physics as it demonstrates that enabling a new way of identifying new features in data using an autoencoder is very achievable using modern FPGAs and can fit the tight timing constraints of collider trigger systems. The authors provide an extensive list of references of usage of neural networks in general to identify new features in data. The results here, which apply to the more constrained environment of online selection, apply to other fields that have high bandwidth data streams and a need to identify anomalies in real-time.

The authors achieve a very compact and fast (30ns) implementation of an autoencoder able to identify a new physics channel (Higgs decay to pseudoscalars, in a different and more challenging final state than in the initial version of this article) with more sensitivity than traditional trigger methods. They also compare an implementation to a previously published implementation that used a different architecture, and for the same FPGA device, same clocking, and same physics samples achieve a faster implementation for similar resource usage.

- Does the work support the conclusions and claims, or is additional evidence needed?

The reported work does support the conclusions. The authors addressed an earlier referee comment questioning the usage of offline precision quantities for their autoencoder to demonstrate sensitivity, and (successfully) make the case that for the HL LHC upgrades of the CMS and ATLAS experiments the precision indeed would be at about that level in their trigger systems.

- Are there any flaws in the data analysis, interpretation and conclusions? - Do these prohibit publication or require revision?

The authors have cleared up some previous reservations about the first version of the article. Aside from their argument justifying use of offline precision variables, they also standardized on a specific FPGA device and clocking frequency to make comparisons across physics models and to previous work easier. Additionally, the authors have made their use-case even stronger by studying a more challenging new physics decay channel to demonstrate that their anomaly-driven approach can achieve factors of 3 to 100 times more sensitivity than traditional cut-based trigger approaches.

- Is the methodology sound? Does the work meet the expected standards in your field?

The work appears sound and is at the expected level of standards for our field.

- Is there enough detail provided in the methods for the work to be reproduced?

Yes, I believe there is enough detail here for reproducibility.

Reviewer #1 (Remarks on code availability):

Actually the link for the code repository requires a login account.

** Thank you for the follow-up. I think there might be a distinction between View Item (login required for authors and not for you to click) and Download (no login required and for you to click). We attach a screenshot of the page to clarify the link. In any case, we have issued a doi for the page. It is <http://dx.doi.org/10.18117/xaw2-9319> . Here is the direct link to the testbench in any case http://d-scholarship.pitt.edu/45784/1/autoencoder_tb_kit_revised.zip .

Reviewer #2 (Remarks to the Author):

The authors have taken into account my concerns. The manuscript has greatly improved.

Reviewer #2 (Remarks on code availability):

1) The provided link (<http://d-scholarship.pitt.edu/id/eprint/45784>) for accessing the code requires a login and a password that I do not have.

** Thank you for the follow-up. I think there might be a distinction between View Item (login required for authors and not for you to click) and Download (no login required and for you to click). We attach a screenshot of the page to clarify the link. In any case, we have issued a doi for the page. It is <http://dx.doi.org/10.18117/xaw2-9319>. Here is the direct link to the testbench in any case http://d-scholarship.pitt.edu/45784/1/autoencoder_tb_kit_revised.zip.

2) Accessing the code through <https://www.fwx.pitt.edu> allows to install the fwMachina framework. The code is documented using Doxygen (with a few minor bugs in the rendering.)

The readme provides all the installation step. It is well explained but the provided method does not work on lxplus (as advertised) but work at ccin2p3. The "hello world" example is working fine. I was not able to find easily how to reproduce the article result (maybe because of 1). A readme in the article folder would have been helpful.

** Please try the testbench above. It has the IP with accompanying README that allows the reproduction of the results. The [fwx.pitt.edu](https://www.fwx.pitt.edu) points to the gitlab repository that holds our earlier version of the decision tree implementation, in which a user can train on their own problem and produce a firmware design. For the method in our paper, we do not provide the general python package to solve the user's own problems, but rather the complete Vivado project along with a testbench to reproduce the results of our paper. If anyone would like to try our general code for their specific problem, we would be happy provide it upon email request as this is outside the scope of the paper. Meanwhile, we will address the README on our webpage in any case.

3) I do not have the Xilinx tools installed so I am not able to test this part

** The version we used can be downloaded at <https://www.xilinx.com/support/download/index.html/content/xilinx/en/downloadNav/vitis/2022-2.html>

Reviewer #3 (Remarks to the Author):

Thank you for taking into account my feedback! I particularly appreciate the new benchmark study. I have two follow up points:

1. Sorry for not being clear about "coordinate dependence". By this I mean if you have your features x and a score that is related to $p(x)$, so you keep events with $p(x) < \text{cut}$, then if $x \rightarrow f(x)$, the events you select need not be the same (even for the same fraction of events selected). This is because there is a Jacobian factor that $p(x)$ picks up when going from x to $f(x)$ and this may not preserve the order of events. This is not a problem per se, but it does limit the interpretability and universality of unsupervised anomaly detection (it is not a challenge just for this paper, but maybe useful to state).

** Thank you for the follow-up. We think we understand your concern! Let's consider a toy example just to be on the same page. Suppose the input variable space is 2-dimensions (x_1 and x_2 ranging from 0 to 1) and the SM training sample is scattered across the diagonal near $x_1 = x_2$. Suppose that the anomalous events are populated near $x_1 = x_2 = 0$ (say for signal-A) as well as $x_1 = x_2 = 1$ (say for signal-B). Then the distance between signal-A and the SM would be roughly the same as for signal-B and the SM. In this case, the symmetry of the scenario gives a two-fold ambiguity in interpreting the events passing a cut on the anomaly score. If we understood correctly, we agree that our construction is not immune to such situations. Even if the bins may not be adjacent (in x_1 - x_2 space), which may be what you're pointing to, we are able to trace-back the anomaly to the two corners of the input space defined by bin boundaries. Regarding the interpretability, our response to your next question may help here.

2. [more importantly] I agree that latency in general is a problem of the HL-LHC, but I remain unconvinced that it is the main problem for anomaly detection. The crux is this: "After the events are saved to disk, full reconstruction of the raw data would be done and a detailed analysis would be conducted. However, all of this is beyond the scope of the paper." What, exactly, do you propose you do with the anomalous events that you save? I am not sure what analysis you will conduct. You need to know how anomalous the events are, which your method does not tell you. One answer to this might be "as a baseline, we compare to simulations". I don't find this very satisfying (after all, we are likely operating in a region where the simulations are not accurate), but at least it would supply one possible use of the data. For offline analyses, background estimation is a critical component. I appreciate that this criticism also applies to Govorkova et al.

** Thank you for the follow-up. We are not proposing to do the offline analysis, but we outline three ideas that might address your concern. We added some text in our revised draft that summarize the following ideas: (1) Record some data in the sideband ($\text{low cut} < x < \text{cut}$) near the anomalous selection ($x > \text{cut}$). Then the corresponding offline analysis can extrapolate the amount of SM background from the data itself as one would do for the traditional mass peak study. (2) Another approach could be to cut on the anomaly score and look at the mass peak, which was not used as inputs, as done in Phys. Rev. Lett. 132 (2024) 8, 081801. We added this reference. (3) Look for self-similar events in the latent space. For our DT AE, we can look at the histogram vs. bin number. Anomalous events from BSM would spike in a specific bin number or group of nearby bin numbers. More generally, one can pursue a more detailed analysis in the latent space similar to Phys. Rev. D 105 (2022) 11, 115009 [2103.06595].

Reviewer #3 (Remarks to the Author):

I appreciate the author's thoughtful reply. A brief followup:

1. For my point about coordinate transformations, I don't understand your response - your first signal lives on the background manifold while the second signal does not (so in what sense are they same distance from the background?). I found this paper which seems to describe the point I'm trying to make: 2209.06225. Like I said, this is not a problem per se of your approach, but it is a weakness of unsupervised anomaly detection in general (there is no optimality guarantee). I'm not sure your new sentences in the last paragraph of Sec. 4 address this, but maybe now my point is clearer.

2. Thank you for your response about background estimation. I understand that you are not proposing to do the offline analysis, but I think it is quite important to comment on it, because saving events is one thing, but being able to use them is another. Like I said before, this is a critique of almost all online anomaly detection protocols, including ones published in this or related journals.

Thank you for adding the additions to the text. It seems like this is the last paragraph of Sec. 4? In my opinion, this paragraph is not strong enough. For example, I did not see any mention of the control region method that you proposed in your written answer. I'm actually not sure it is as simple as you say - most (all?) control region extrapolations are done on non-ML observables. When you use a neural network, you are extrapolating, and what's to say that the data in the control region actually constrains the performance in the signal region? Also, with unsupervised ML, it is very possible that the signal is also in the control region (you can't know ahead of time). I'm not proposing that you solve this, but a paragraph where you describe this challenge would be necessary. This would already be more than what many of the other papers in this area have done!

If you could modify/extend that last paragraph a bit, then I could recommend publication.

Reviewer #3 (Remarks to the Author):

I appreciate the author's thoughtful reply. A brief followup:

1. For my point about coordinate transformations, I don't understand your response - your first signal lives on the background manifold while the second signal does not (so in what sense are they same distance from the background?). I found this paper which seems to describe the point I'm trying to make: 2209.06225. Like I said, this is not a problem per se of your approach, but it is a weakness of unsupervised anomaly detection in general (there is no optimality guarantee). I'm not sure your new sentences in the last paragraph of Sec. 4 address this, but maybe now my point is clearer.

**** Thank you for the clarification. We read the paper and now understand your point. We cite this paper in our revised version. We added text to clarify this point that rarity depends on the choice of variables.**

2. Thank you for your response about background estimation. I understand that you are not proposing to do the offline analysis, but I think it is quite important to comment on it, because saving events is one thing, but being able to use them is another. Like I said before, this is a critique of almost all online anomaly detection protocols, including ones published in this or related journals.

Thank you for adding the additions to the text. It seems like this is the last paragraph of Sec. 4? In my opinion, this paragraph is not strong enough. For example, I did not see any mention of the control region method that you proposed in your written answer. I'm actually not sure it is as simple as you say - most (all?) control region extrapolations are done on non-ML observables. When you use a neural network, you are extrapolating, and what's to say that the data in the control region actually constrains the performance in the signal region? Also, with unsupervised ML, it is very possible that the signal is also in the control region (you can't know ahead of time). I'm not proposing that you solve this, but a paragraph where you describe this challenge would be necessary. This would already be more than what many of the other papers in this area have done! If you could modify/extend that last paragraph a bit, then I could recommend publication.

**** Thank you for the follow-up. We agree with your assessment. As a result, we state the challenge in how to analyze the selected events. For the latter, we list three ideas in the literature. (1) Bump hunt, as done by ATLAS. (2) Use of sidebands and added two new citations. (3) Study of the latent space. Lastly, since this journal is broader than high energy physics, we added a sentence on the need for a simultaneous fit with various subsamples.**